# Towards inferring nanopore sequencing ionic currents from nucleotide chemical structures

Hongxu Ding [1,2,3 ✉], Ioannis Anastopoulos [1,2,3], Andrew D. Bailey IV [1,2,3], Joshua Stuart [1,2 ✉] & Benedict Paten [1,2 ✉]

The characteristic ionic currents of nucleotide kmers are commonly used in analyzing nanopore sequencing readouts. We present a graph convolutional network-based deep learning framework for predicting kmer characteristic ionic currents from corresponding chemical structures. We show such a framework can generalize the chemical information of the 5-methyl group from thymine to cytosine by correctly predicting 5-methylcytosine-containing DNA 6mers, thus shedding light on the de novo detection of nucleotide modifications.

[1] Department of Biomolecular Engineering, UC Santa Cruz, Santa Cruz, CA, USA. [2] UC Santa Cruz Genomics Institute, Santa Cruz, CA, USA. [3] These authors contributed equally: Hongxu Ding, Ioannis Anastopoulos, Andrew D. Bailey IV. ✉email: hding16@ucsc.edu; jstuart@ucsc.edu; bpaten@ucsc.edu

During nanopore sequencing, consecutive nucleotide sequence kmers block the pores sequentially, producing ionic currents[1]. Chemical modifications on nucleotides additionally alter the ionic currents measured during nanopore sequencing[2–22]. The characteristic ionic currents of kmers, which are represented in kmer models, are used in interpreting nucleotide modifications[2,3,7,23]. Up to now, 29[2–11] and 30[12–22] modifications have been successfully characterized in DNA and RNA, respectively.

To date, most modification analysis algorithms are based on kmer models[2,3,7,23]. However, such learning strategies struggle to generalize knowledge between related kmers. For example, our previous hierarchical Dirichlet process approach could be structured to learn associations between kmers with specific shared properties, e.g., by numbers of pyrimidine bases, but could not generally learn relationships between arbitrary chemical similarities[2]. Moreover, such approaches necessarily represent base modifications as distinct, unrelated characters. The upshot is that such kmer character-based models require extensive training data and are unable to de novo predict the impact of a chemical modification. Given that the number of possible kmers increases polynomially with the number of modifications being modeled, it is extremely challenging to generate sufficient control data for such models, especially considering that more than 50 and 160 nucleotide modifications have been verified in DNA and RNA, respectively[24,25].

To start to tackle this problem, we propose a graph convolutional network (GCN)-based deep learning framework[26,27] for predicting kmer characteristic ionic currents from corresponding kmer chemical structures. We confirm that the proposed framework is able to represent individual kmer chemical modules, such as the phosphate group and the sugar backbone, as well as the nucleobase methyl and amine groups. We further demonstrate that this framework can infer full kmer models even when the training data does not include all possible kmers. This opens up the possibility of modeling kmers that are under-represented in control datasets. We also show that the framework can generalize the 5-methyl group in thymine to cytosine, thereby accurately predicting the characteristic ionic currents of 5-methylcytosine (5mC)-containing DNA 6mers. Such generalization of chemical information is a reason for optimism about the potential for de novo detection of nucleotide modifications.

## Results

**Architecture of the deep learning framework**. Our deep learning framework consists of three groups of layers, including GCN layers, convolutional neural network (CNN) layers, and one fully connected neural network (NN) layer. As shown in Fig. 1A, the kmer chemical structures are first represented as graphs, with atoms as nodes and covalent bonds as edges. The atom chemical properties are then assigned as node attributes. Based on such graphs, GCN layers extract one chemical feature vector for every atom, by visiting its immediate graph neighbors. By this means, after several GCN layers, atom feature vectors will contain chemical information for all atoms within a certain graph distance. Specifically, this distance equals the number of GCN layers applied. Considering the small encoding distance of each layer of a GCN, to improve the encoding efficiency of the framework, CNN layers are then applied to summarize relatively long-range chemical information above the GCN layers. The output matrices of the final CNN layer are then "flattened" as feature vectors. Such feature vectors are then passed to the final fully connected NN layer to summarize kmer-level information and finally predict the kmer characteristic ionic currents (see "Methods"). For DNA and RNA, the corresponding best-performing architecture

in hyperparameter tuning was selected for downstream analysis (see "Methods").

**Kmer-level generalization**. We first confirmed that the proposed framework can accurately predict characteristic ionic currents of kmers from their chemical structures. To do so, we performed a downsample analysis on the canonical DNA 6mer model provided by Oxford Nanopore Technologies (ONT, see "Methods"), by randomly partitioning canonical DNA 6mers with various train-test splits. For each train-test split group, we performed 50-fold cross-validation and used root mean square error (RMSE) and Pearson's correlation ($r$) to quantify the goodness of fit (see "Methods"). As shown in Fig. 1B, Supplementary Fig. 1, and Supplementary Table 1, the performance stabilized as more than 40% of DNA 6mers were included in the training. Specifically, for DNA 6mers only used in the test, average RMSE and Pearson's correlation reached 1 and 0.995, respectively. Such a result indicated on average 40% of randomly selected DNA 6mers contain sufficient information to recapitulate the full DNA 6mer model.

We next explored how specific kmer training subsets influence the ionic current predictions. Specifically, we trained the framework using either the DNA 6mers that (a) do not contain a given nucleotide (base dropout), (b) do not specify a nucleotide at a given position (position dropout), or (c) that are combined from different base dropouts (for instance, using the union of A-dropout and T-dropout kmers, such that kmers containing both A and T would be excluded, but not kmers containing either A or T, noted as A–T model combination, see "Methods" and Supplementary Note 1 for details). This latter combination analysis simulates the situation in which we have knowledge about two modifications independently, but must guess at the effect of their combination. For each group in (a–c), 50 independent repeats were performed, and goodness of fit was used to evaluate the performance. As shown in Fig. 1B and Supplementary Fig. 1, base and position dropouts significantly decreased the prediction power. Moreover, dropouts in third and fourth positions contributed the most to the decrease in prediction power, followed by the second and fifth positions, consistent with prior observations[28]. Model combinations, on the other hand, in general, had a minor influence on the prediction power.

The above-mentioned analyses together suggest, once properly trained with sufficient and diverse 6mers, the kmer-level generalizability of the framework. To further validate and extend our framework, we performed all the above-mentioned analyses using RNA, switching to using 5mers instead of 6mers to match the available training data. Considering the significantly smaller amount of training data (1/4th the number of distinct RNA 5mers vs DNA 6mers), the prediction power of the RNA architecture is compromised. However, once trained with a similar number of kmers, the RNA architecture yielded comparable prediction power. For instance, the RNA 0.95–0.05 (972 training kmers) and DNA 0.25–0.75 (1024 training kmers) train-test splits yielded comparable performance on test data. Such a result suggests the validity of our proposed architecture (see "Methods," Supplementary Fig. 2, and Supplementary Note 2 for details).

Such kmer-level generalizability could facilitate nucleotide modification detection by greatly reducing the required control data to generate reliable full modification-containing kmer models. As a proof of concept, we trained the DNA deep learning architecture with all canonical 6mers plus {1%, 5%, 10%, 30%, 50%, 70%, 90%} of randomly selected 5mC-containing 6mers ("modification imputation" analysis). The characteristic ionic current signals of such 5mC-containing DNA 6mers were

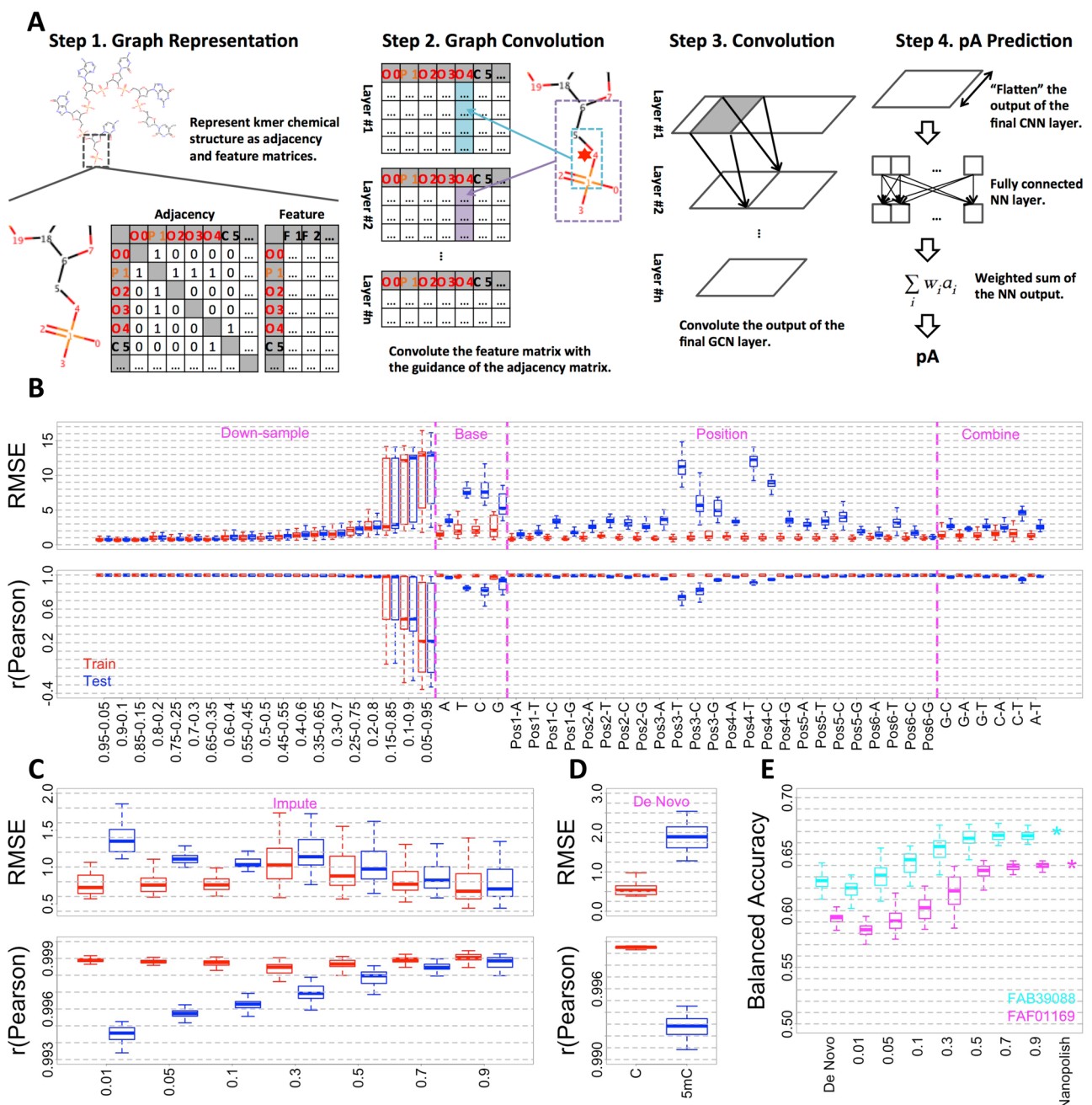

**Fig. 1 Predicting kmer characteristic ionic currents from chemical structures. A** Graphic overview of the proposed deep learning framework for DNA analysis. **B** Goodness of fit of DNA canonical random downsample, base-dropout, position-dropout, and model combination analyses. Specifically, "downsample" denotes the random dropout experiment, where we create random train-test splits. "Base" denotes base-dropout experiment, where we drop DNA 6mers that contain a specific base in any given position during training. "Position" denotes positional base-dropout experiment, where we drop DNA 6mers that contain a specific base in a given position during training. As for "combine," we drop DNA 6mers that contain both of the specified bases during training. **C** Goodness of fit of 5mC-containing DNA 6mer imputation analysis. **D** Goodness of fit of de novo 5mC-containing DNA 6mer prediction. C and 5mC refer to the goodness of fit of canonical DNA 6mers and 5mC-containing DNA 6mers, respectively. In **B**–**D**, Train (red) and Test (blue) refer to the goodness of fit of the training and test DNA 6mers, respectively. **E** Predictive accuracy of C/5mC status quantified by balanced accuracy. Nanopolish, predictive analysis with the nanopolish model as baseline control. De Novo, predictive analysis with 5mC-containing DNA 6mer models described in (**D**), which were predicted from canonical training. 0.01–0.9, predictive analysis with different imputation 5mC-containing DNA 6mer models as described in (**C**). FAB39088 (cyan) and FAF01164 (purple) refer to two independent NA12878 cell line native genomic DNA nanopore sequencing datasets. Throughout (**B**–**E**), the median, minimum/maximum (excluding outliers), and first/third quartile values were shown by the boxplots.

obtained from the nanopolish model as reported in refs. [3,23]. For each training group, 50 independent repeats were performed (see "Methods"). As shown in Fig. 1C and Supplementary Fig. 3, decent goodness of fit could be obtained when as few as 5% of 5mC-containing DNA 6mers were used as training data.

Specifically, for test DNA 6mers, the average RMSE and Pearson's correlation reached 1.2 and 0.995, respectively. Furthermore, models trained with the knowledge of 50% 5mC-containing DNA 6mers performed about as well as models trained with 90%.

**Chemical group-level generalization in DNA 5mC de novo prediction**. We noted that performance of the model on held out 5mC kmers trained with just 1% of 5mC kmers was better than chance. This raised the question of if chemical group-level information was being usefully generalized among nucleotides by our framework, potentially allowing the 5mC to be predicted de novo, without ever having been seen by the model. As a chemical derivative of cytosine, 5mC contains an additional methyl group at the fifth position (5-methyl) of the pyrimidine ring. This 5-methyl group is shared between 5mC and thymine. We thus hypothesized that 5mC can be generalized by combining the pyrimidine ring from cytosine and 5-methyl group from thymine. As a proof of concept, we trained the framework with all canonical DNA 6mers to make de novo predictions on 5mC-containing DNA 6mers. Similar to previous analyses, 50 independent repeats were performed, and the prediction power was first quantified by goodness of fit against the above-mentioned nanopolish model. As shown in Fig. 1D and Supplementary Fig. 3, although goodness of fit of 5mC-containing DNA 6mers was significantly worse than the canonical counterparts, decent performance could still be obtained (average RMSE and Pearson's correlation reached 1.8 and 0.993, respectively). We also compared the goodness of fit between canonical and 5mC-containing DNA 6mers, and as shown in Supplementary Fig. 4, a positive correlation trend could be observed. Such a result confirmed that no overfitting was introduced during architecture training with canonical DNA 6mers, and further suggested 5-methyl generalization.

**Human genome C/5mC-status predictive analysis**. We next performed "predictive analysis" to test whether the DNA 6mer models inferred by our deep learning framework could be used to correctly predict DNA C/5mC status at a per-read, per-site resolution from ionic currents ("predictive accuracy," see "Methods"). C/5mC sites to be predicted were confirmed by bisulfite sequencing (see "Methods"). We also quantified the predictive accuracy with the above-mentioned nanopolish model as a baseline control (see "Methods"). As shown in Fig. 1E, average predictive accuracy, quantified by balanced accuracy (BA), became comparable with baseline control with 50% of imputed 5mC-containing 6mers. Taken together, these results confirmed the kmer-level generalizability of our framework, as well as suggesting that reliable modification-containing kmer models can be built with significantly less control data once facilitated by our methodology. Such a result confirmed the successful 5-methyl generalization. More confusion matrix-based prediction evaluations can be found in Supplementary Fig. 5.

**The encoding of chemical structures**. To better understand how chemical structures were encoded, we visualized DNA 6mer atom similarity matrices. Specifically, we trained the proposed framework with all canonical DNA 6mers. We then calculated and visualized the Pearson's correlations of the feature vectors derived by the final GCN layer as atom-level similarities. As shown in Supplementary Fig. 6, we visualized ten randomly chosen canonical DNA 6mers. Taking CGACGT as an example, as shown in Fig. 2A, C, atoms were in general aggregated by chemical contexts. For instance, as shown in (A), for the first cytidine monophosphate in CGACGT, atoms #0–4 were tightly clustered with average $r > 0.9$, recapitulating the phosphate group. Atoms #5–8 and #17–18 are also clustered with average $r > 0.9$, denoting the deoxyribose backbone. Among cytosine atoms #9–16, #9 nitrogen atom connected the nucleobase to the deoxyribose backbone, atoms #10–11 denoted the C=O group, and atoms #12–16 composed the C=C-C=N conjugation system and the covalently bonded amine group. Similarly, atoms in other

nucleotides can also be clustered into phosphate groups, deoxyribose backbones and nucleobases. Within the nucleobases, chemical modules including chemical groups and conjugation systems can further be dissected. Such a phosphate-deoxyribose-nucleobase pattern repeated and constituted DNA 6mers.

We also examined the inter-nucleotide similarities of different components. As shown in Fig. 2A, C, in general high similarities (average $r > 0.9$) were observed among phosphates, as well as deoxyriboses from different nucleotides. Meanwhile, chemical modules sharing similar structures, e.g., the conjugation systems of adenines, cytosines, and guanines were more similar to each other. On the other hand, low similarities (average $r < 0.5$) were observed between chemical modules with distinct structures, e.g., the cytosine C=O group and the thymine methyl group. Taken together, these results suggest that the GCN layers in the proposed framework can effectively capture features interpretable as individual chemical modules.

We further visualized the atom-level similarity matrices of 5mC-containing DNA 6mers, aiming to understand the generalization of methyl group among thymine and 5mC. We thus trained our deep learning framework with all canonical DNA 6mers, calculated the Pearson's correlations of the feature vectors derived by the final GCN layer, and further visualized such atom-level similarity matrices of ten randomly selected 5mC-containing DNA 6mers (Supplementary Fig. 7). Taking GT(5mC)AGA as an example (Fig. 2D, F), the phosphate-deoxyribose-nucleobase repetitive pattern was recapitulated. Within nucleobases, high similarities (average $r > 0.9$) were again observed among chemical modules with similar structures. Specifically, strong similarities (average $r > 0.9$) were observed between thymine (#38) and 5mC (#58) methyl groups (Me). In addition, such methyl groups were uniquely encoded as they were less correlated with any other DNA 6mer chemical modules (average $r < 0.5$). We also quantified the atom-level similarity between GT(5mC)AGA and corresponding canonical counterpart GTCAGA. As shown in Supplementary Fig. 8, strong similarities (average $r > 0.9$) were observed between GT(5mC)AGA and GTCAGA thymine methyl groups, as well as the 5mC-methyl groups from GT(5mC)AGA and thymine methyl groups from GTCAGA. These observations together suggested the successful chemical information generalization. Noticeably, the methyl groups were encoded with the pyrimidine backbone C=C modules. Such a result suggests that the GCN encoding is driven by chemical context, which further implies when generalizing one specific chemical group among different nucleotides, the corresponding chemical contexts in which such chemical group resides should be the same.

Finally, we projected kmer atom feature vectors into the tSNE space, in order to summarize the atom-level similarity matrices further providing a global visualization of kmer atoms. As shown in Fig. 2B, E, atoms under the same chemical context clustered together, e.g., phosphate group phosphate atoms (#1, #20, #42, #63, #82, and #104 in B and #1, #23, #43, #63, #84, and #106 in F) and deoxyribose ring oxygen atoms (#7, #26, #48, #69, #88, and #110 in B and #7, #29, #49, #69, #90, and #112 in E), as well as NH₃ group nitrogen atoms (#14, #35, #55, #76, and #97 in B and #16, #56, #76, #99, and #119 in E). Specifically, as shown in E, in 5mC-containing DNA 6mer GT(5mC)AGA, T-methyl group carbon atom #38 and 5mC-methyl group carbon atom #58 clustered together, along with pyrimidine backbone C=C module atoms #37 and #39 in T, as well as #57 and #59 in 5mC. Taken together, these results confirm that GCN could properly encode chemical structures based on the corresponding chemical contexts.

**Analyzing the 2mG site in *Escherichia coli* 16S ribosomal RNA (rRNA)**. Our deep learning framework could potentially shed

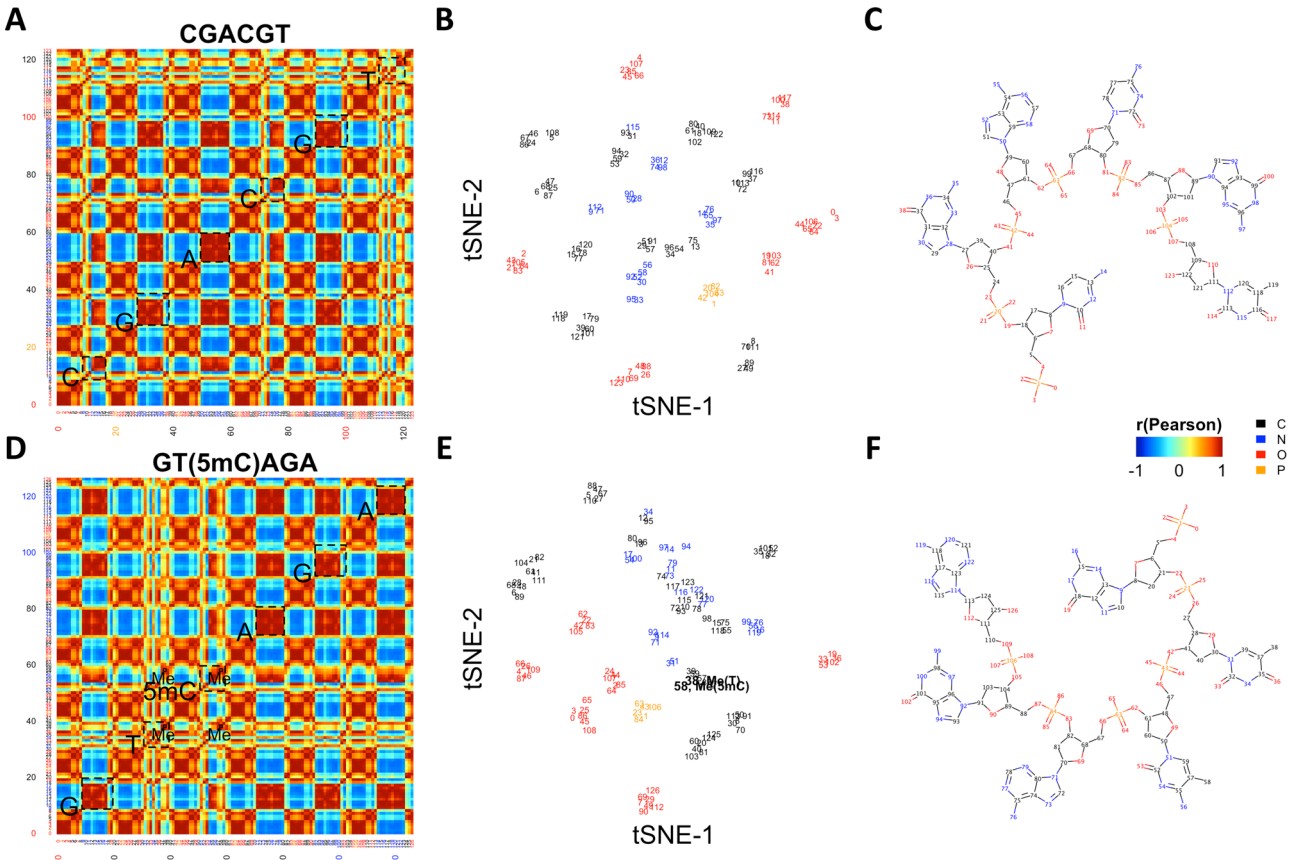

**Fig. 2 Visualizing the encoding of chemical structures. A–C** Atom similarity matrix, tSNE visualization, and chemical structure of the example canonical DNA 6mer CGACGT. In **A**, **B**, atoms were numbered and colored based on the chemical structure in (**C**). Carbon, nitrogen, oxygen, and phosphorus were colored as black, blue, red, and orange, respectively. Specifically, in **A**, nucleobases were highlighted by dashed boxes. **D–F** Atom similarity matrix, tSNE visualization, and chemical structure of the example 5mC-containing DNA 6mer GT(5mC)AGA. In **D**, **E**, atoms were numbered and colored based on the chemical structure in (**F**). Carbon, nitrogen, oxygen, and phosphorus were colored as black, blue, red, and orange, respectively. Specifically, in **D**, **E**, methyl group carbon atoms (#38 in T and #58 in 5mC) were highlighted.

light on previously understudied, less prevalent nucleotide modifications. As a proof of concept, we analyzed 2mG, which can be represented as the purine ring in guanine with the N2-methyl group in 6mA. Specifically, we generated an RNA 5mer model using canonical and 6mA-containing kmers (see "Methods"). We then predicted the characteristic ionic current signals of 2mG-containing RNA 5mers (see "Methods"). To test our predictions, we analyzed nanopore sequencing reads of *E. coli* 16S rRNA transcript J01859.1, which contains an annotated 2mG at position 1206 (see "Methods"). As shown in Supplementary Fig. 9, our predictions recapitulated the characteristic ionic current signals of 2mG-containing and pairing canonical RNA 5mers (see "Methods"). Moreover, we confirmed that such predicted characteristic ionic current signals could be used to correctly determine the G/2mG modification status (see "Methods").

## Discussion

We propose a GCN-based deep learning framework for associating kmer chemical structures with corresponding characteristic ionic currents. We show that such a framework can recapitulate full kmer models from partial training data, thus greatly facilitating modification analysis by reducing the amount of required control data. Specifically, for cases where a small proportion of random kmers are under-represented in control data, we can apply the same principle as the downsample analysis to learn around these training deficiencies. For cases where comprehensive control datasets are available only for single

modifications, we could apply model combination (as we showed for individual nucleotides) to model kmers containing multiple modifications simultaneously.

We further demonstrated that our framework can represent novel modifications by generalizing encoded chemical groups between nucleotides, thus shedding light on de novo modification detection. However, the current model is not without its limitations. For example, the proposed framework encodes chemical groups, e.g., the methyl groups in thymine and 5mC, as well as the amine groups in cytosine, guanine, and adenine, with covalently bonded "backbone atoms," showing a strong chemical context-specificity (Fig. 2 and Supplementary Figs. 6 and 7). Thus, the current framework cannot properly handle "stacked" chemical groups. For instance, the methylamine group in N6-methyladenine (6mA) cannot be correctly encoded by simply stacking methyl with an amine. As shown in Supplementary Fig. 10, substituting A with 6mA was predicted to decrease characteristic ionic currents, which is the opposite of a previous study[6]. Therefore, the extensibility of the framework is currently limited. To overcome such a limitation, controlled nanopore sequencing profiles of diverse nucleotide modifications are needed, in addition to the modeling of other chemical interactions.

Deep learning-based approaches have emerged as powerful tools for detecting nucleotide modifications from nanopore sequencing readouts. Compared to kmer model-based counterparts, deep learning-based approaches are reported to have better accuracy and less computational resource consumption[5,8].

Recently, ONT released the megalodon algorithm (https://github.com/nanoporetech/megalodon), which can drastically increase the accuracy for 5mC identification (Supplementary Fig. 5, see "Methods"). Thus, one potential future extension of the paper would be using the learned models as components of a larger, recurrent deep NN.

Another potential future direction would be generalizing the proposed framework to handle both DNA and RNA kmers. Due to different translocation speeds, the nanopore sequencing ionic currents of DNA and RNA are not directly comparable[29]. Therefore, advanced deep learning frameworks, which can take both kmer chemical structures and nanopore sequencing experimental setups, are needed. Considering DNA and RNA share several noncanonical nucleobases, e.g., inosine (I)[30], we might combine the ribose in RNA and I in DNA to reconstruct I-containing RNA 5mers, and vice versa for I-containing DNA 6mers. By this means, required RNA control nanopore sequencing reads, which are usually challenging to obtain, can be largely compensated. Meanwhile, such generalization would largely diversify the chemical contexts that can be represented, further facilitating the de novo modification analysis.

## Methods

**Graph representation of kmer chemical structures**. Following the workflow described in ref. [26], kmer chemical structures were first described by SMILES (Simplified Molecular Input Line Entry System) strings, which were assembled by concatenating SMILES strings of individual nucleotides, as summarized in Table 1. Each nucleotide base can be described by several SMILES strings. The SMILES strings presented in Table 1 were selected due to the ease of combining them into complete kmers. Based on the information provided by ONT, as well as a previous study[28], DNA and RNA are represented by 6mer and 5mer, respectively. An "O" was then added to the end of each concatenation to represent the residual unbonded hydroxyl group on the sugar backbone.

We then represent the SMILES string of each kmer as a graph noted as $G(A, X)$. Specifically, the topology (atom order is determined by SMILES string) of each kmer chemical structure was represented by an adjacency matrix $A$, with $A_{i,j}$ equals 1 iff the $i$th and $j$th atoms were covalently bonded. Meanwhile, for every atom in $A$, the corresponding chemical properties were represented by feature matrix $X$, with $X_i$ recording the chemical property vector for the $i$th atom. Atom chemical properties included in the study were summarized in Table 2.

Therefore, the GCN has encoded input a chemical feature matrix $X$ with the guide of chemical topology matrix $A$, representing kmer chemical structures. Notably, for convenient GCN implementation, the size of $A$ and $X$ is kept constant. Due to the variable number of atoms across kmers, $A$ and $X$ were thereby padded with zeros based on the largest kmers. Specifically, the $A$ matrix was padded at the end of its rows and columns, with dim($A$) is {133, 133} and {116, 116} for DNA and RNA, respectively. While the $X$ matrix was padded at the end of its rows, with dim($X$) is {133, 8} and {116, 8} for DNA and RNA, respectively. Note that the kmer representation is guided by the nonzero elements (covalent bonds) in $A$, thus such padding will not affect the GCN encoding.

**Architecture of the deep learning framework**. The GCN layers of our framework were built based on the procedure described by ref. [26]. Fast approximate convolutions on $G$ were used to create a graph-based NN $f(X, A)$, following the propagation rule:

$$H^{(l+1)} = \sigma(\tilde{U}^{\frac{1}{2}} \tilde{A} \tilde{U}^{\frac{1}{2}} H^{(l)} W^{(l)})$$

$\sigma(\bullet)$ is the activation function applied to each layer. Here, the activation function used was the exponential linear unit (ELU). $\tilde{U}_{i,j} = \sum_j A_{i,j}$ is the degree matrix for each atom in the graph. $\tilde{A} = A + I$ adds self edges to each of the atoms. The $\tilde{U}^{\frac{1}{2}} \tilde{A} \tilde{U}^{\frac{1}{2}}$ transformation prevents changes in the scale of the feature vectors[26] and constructs filters for the averaging of neighboring node features. $H$ and $W$ denote the output (activation vectors) and weights of each GCN layer, respectively. The corresponding superscript represents the layer index. $H^0 = X$; however, subsequent $H$ represents the GCN-derived features.

The intuition of the graph convolution process is described as follows. For every kmer, chemical properties of atoms, together with their covalently bonded neighbors, will be convoluted with the guidance of $G$. Such graph convolution yields an activation matrix $H$, following the aforementioned propagation rule. $H$ is an atom-by-feature matrix, with dimensions {133, N} and {116, N} for each of the DNA and RNA kmers, respectively. Here, N equals the number of nodes of the GCN layer, which determines the number of features to be derived. The selection rule for N is described in the following section. As more GCN layers are stacked, the graph convolution process is repeated. The $H$ matrix will thus contain chemical information of all atoms within a certain graph distance, which equals the number of GCN layers applied. By this means, "chemical modules" composed of several atoms linked by covalent bonds are encoded.

Considering the small encoding distance of a GCN, for a better encoding efficiency we wanted additional layers that can quickly summarize atom information. We applied standard 1-D CNN layers with rectified linear unit activation right after the GCN layers. Average Pooling[31] was applied on the output of each 1-D CNN layer. Average Pooling takes the average of each $2 \times 2$ patch of the CNN output matrix. Specifically, the output dimension of the first CNN layer equals $\{133 - K + 1, N'\}$ and $\{116 - K + 1, N'\}$ for DNA and RNA kmers, respectively. Here, $K$ is the CNN kernel size and $N'$ is the node number of the final GCN layer. Output dimensions of subsequent CNN layers equals $\{m - K + 1 - 2 + 1, n - 2 + 1\}$, where $\{m, n\}$ denotes the output dimension of the previous layer and 2 denotes the Average Pooling patch size. The output from the final 1-D CNN layer, after Average Pooling, was passed to a Flatten layer, which converts the final 1-D CNN output matrix to a 1-D feature vector in a row-wise fashion. The NN layer then takes the flattened vector as input, thereby summarizing information about the entire kmer and producing a highly informative representation. Elements of the NN layer output vector are linearly combined as the final pA value.

**Training procedure**. Our framework was trained with the Keras[32] framework (2.3.1) with TensorFlow backend using the Adam[33] optimizer for gradient descent optimization. The framework was allowed to train for a maximum of 500 epochs. To control for overfitting, EarlyStopping[34] was used by monitoring the increase in validation loss. Early termination of training was reached if the validation loss was increasing for ten consecutive epochs, indicating that the framework had reached maximum convergence. A mean-squared error was used as the loss function during the training process. Meanwhile, a 10% random dropout was applied after each layer, to further prevent overfitting[35]. In the following experiments, the exact same training routine was used.

**Hyperparameter tuning**. In order to determine the optimal architecture, we performed a hyperparameter grid search. The search involved the hyperparameters shown in Table 3.

We used the following scaling factor to determine the number of nodes in each GCN/CNN layer of our framework:

$$n = 16 \times 2^{(l-1)},$$

| Table 1 SMILES strings of individual nucleotides. | |
|---|---|
| **Nucleotide** | **SMILES string** |
| A (DNA) | OP(=O)(O)OCC1OC(N3C=NC2=C(N)N=CN=C23)CC1 |
| T (DNA) | OP(=O)(O)OCC1OC(N2C(=O)NC(=O)C(C)=C2)CC1 |
| C (DNA) | OP(=O)(O)OCC1OC(N2C(=O)N=C(N)C=C2)CC1 |
| G (DNA) | OP(=O)(O)OCC1OC(N2C=NC3=C2N=C(N)NC3=O)CC1 |
| 5mC (DNA) | OP(=O)(O)OCC1OC(N2C(=O)N=C(N)C(C)=C2)CC1 |
| 6mA (DNA) | OP(=O)(O)OCC1OC(N3C=NC2=C(NC)N=CN=C23)CC1 |
| A (RNA) | OP(=O)(O)OCC1OC(N3C=NC2=C(N)N=CN=C23)C(O)C1 |
| U (RNA) | OP(=O)(O)OCC1OC(N2C(=O)NC(=O)C=C2)C(O)C1 |
| C (RNA) | OP(=O)(O)OCC1OC(N2C(=O)N=C(N)C=C2)C(O)C1 |
| G (RNA) | OP(=O)(O)OCC1OC(N2C=NC3=C2N=C(N)NC3=O)C(O)C1 |
| 6mA (RNA) | OP(=O)(O)OCC1OC(N3C=NC2=C(NC)N=CN=C23)C(O)C1 |
| 2mG (RNA) | OP(=O)(O)OCC1OC(N2C=NC3=C2N=C(NC)NC3=O)C(O)C1 |

**Table 2 Atom chemical properties included in the study.**

| Feature | Description |
|---|---|
| Carbon | 1 if the atom is carbon, 0 otherwise (boolean) |
| Nitrogen | 1 if the atom is nitrogen, 0 otherwise (boolean) |
| Oxygen | 1 if the atom is oxygen, 0 otherwise (boolean) |
| Phosphorus | 1 if the atom is phosphorus, 0 otherwise (boolean) |
| Atom degree | Total number of covalent bonds around an atom (integer) |
| Implicit valence | It equals the valence of the atom minus the valence calculated from the bond connections (integer) |
| Number of hydrogens | Total count of hydrogens (integer) |
| Aromaticity | 1 if atom in an aromatic ring, 0 otherwise (boolean) |

**Table 3 Hyperparameters searched in the study.**

| Parameters | Space searched | ATGC DNA | AUGC RNA | A(6mA)UGC RNA |
|---|---|---|---|---|
| The number of GCN layers | {2, 3, 4, 5, 6} | 4 | 4 | 6 |
| The number of CNN layers | {2, 3, 4, 5, 6} | 3 | 5 | 6 |
| The kernel size for the CNN layers | {2, 4, 10, 20} | 10 | 10 | 10 |
| The number of nodes in the dense (NN) layer | {32, 128, 512, 2048, 8192} | 8192 | 8192 | 8192 |

where $l$ is the layer index of the GCN, CNN, and NN layer groups. For instance, the number of GCN layers determined to yield the best performance for DNA was 4. The number of nodes for each GCN layer was therefore 128, 64, 32, and 16. The same logic was applied to all other layer groups.

We performed 10-fold cross-validation for each hyperparameter combination. The combination that produced the lowest average RMSE across all folds was adopted as the optimal architecture. The optimal framework for DNA analysis (ATGC DNA) has four GCN layers and three CNN layers with a kernel size of 10 and 8192 nodes in the NN layer. The optimal framework for canonical RNA analysis (AUGC RNA) has four GCN layers and five CNN layers with a kernel size of 10 and 8192 nodes in the NN layer. The optimal framework for modified RNA analysis (A(6mA)UGC RNA) has six GCN layers and six CNN layers with a kernel size of 10 and 8192 nodes in the NN layer.

**Downsample, base-dropout, position-dropout, and combination analysis.** For downsample analysis, we performed random train-test splits in 5% intervals, noted as 0.95–0.05, etc. For base-dropout analysis, we created training sets by removing certain bases. Such train-test split creates 729/4096 (18%) training kmers and 3367/4096 (82%) test kmers for DNA and 243/1024 (24%) training kmers and 781/1024 (76%) test kmers for RNA. It is important to note that everytime a base is dropped from the training set, it is retained in the test set. Similar to base dropout, the position dropout adds one more dimension, which is the position of the nucleotide base. For a given position dropout, the testing kmers are all kmers with the dropout nucleotide covering the target position, and the training kmers are the remaining kmers. Such position dropout creates 3072/4096 (75%) training kmers and 1024/4096 (25%) test kmers for DNA and 768/1024 (24%) training kmers and 256/1024 (25%) test kmers for RNA. It is important to note that bases dropped in a specific position in the training appear in the same position in testing. For combination analysis, we combine the framework by combining any of the two base-dropout kmer sets. For instance, all G- and C-dropout DNA 6mers were noted as G–C. Such analysis creates 1394/4096 (34%) training kmers and 2702/4096 (66%) test kmers for DNA and 454/1024 (44%) training kmers and 570/1024 (56%) test kmers for RNA. For each above-mentioned train-test split, in order to perform statistical analyses, we produced 50 independently trained frameworks for each experiment. Specifically, we performed 50-fold cross-validation in the downsample analysis, considering for each fold the train kmers were randomly selected. As for other analyses, we performed 50 independent repeats using the same training kmer sets. The variability among repeats came from the stochasticity of the training process. To confirm the robustness of our architecture, we further performed two independent replicates (Run-1 and Run-2) of 50.

**Predicting modification-containing kmers.** For the 5mC imputation experiment, the framework was trained on all 4096 {A, T, C, G} DNA 6mers plus {1%, 5%, 10%, 30%, 50%, 70%, 90%} of randomly selected 5mC-containing DNA 6mers, following the training process as described above. In order to perform statistical analyses, we produced 50 independently trained frameworks (50 independent repeats) for each category, with a total of two independent replicates (Run-1 and Run-2) of 50. Such frameworks were then applied on all 15,625 possible {A, T, C, G, 5mC} DNA 6mers.

For the chemical group-level generalization experiment, the framework was trained on all 4096 {A, T, C, G} DNA 6mers following the training process as described above. In order to perform statistical analyses, we produced 50 independently trained frameworks (50 independent repeats), with a total of two

independent replicates (Run-1 and Run-2) of 50. Such frameworks were then applied on all 15,625 possible DNA 6mers, including those composed of {A, T, C, G, 5mC} and {A, T, C, G, 6mA}.

For the 2mG prediction experiment, the framework was trained by the union of {A, U, C, G} and {6mA, U, G, C} RNA 5mers (GSE124309 model, in total 1805 RNA 5mers), which were reported in ref. [13], following the training process as described above. In order to perform statistical analyses, we produced 50 independently trained frameworks (50 independent repeats). Such frameworks were then applied on all 7776 possible {A, 6mA, U, G, 2mG, C} RNA 5mers.

**Human genome C/5mC-status predictive analysis**

*Overview.* To test whether the predicted {A, T, G, C, 5mC} DNA 6mer models can be used to correctly interpret C/5mC status from nanopore readouts, we performed predictive analysis by using signalAlign to make per-read per-base predictions[2]. For a given reference position, signalAlign can produce posterior probabilities for all possible bases based on a provided kmer model. Thus, for DNA 6mer models generated as described in "predicting modification-containing kmers," the empirical nanopolish[3,23] model obtained as described in "kmer models," we allowed signalAlign to predict between C and 5mC. Considering no significant goodness-of-fit differences were observed between Run-1 and 2, only models generated in Run-1 were used here. All predictive analyses performed in this paper were within the human NA12878 cell line.

*Selecting prediction sites.* The prediction sites were selected among the entire human genome. To avoid artifacts caused by ambiguous genomic DNA modification status, we only focused on confident 5mC sites and canonical genomic regions in our analysis. Besides 5mC, other modifications exist in genomic DNA. Considering extremely low fractions of other modifications, e.g., only ~0.05% are modified as 6mAs in the human genome[36], we define "non-5mC" sites as "canonical regions" during predictive analysis. Among these canonical regions, we used the Poisson process with lambda equals 50 to randomly select genomic sites for signalAlign to predict. Such selected sites were at least 12 nucleotides apart, avoiding potential interference by the neighbors. We thus obtained confident 5mC and C sites for signalAlign prediction.

The genomic DNA C/5mC status was determined by analyzing two independent NA12878 cell line bisulfite sequencing datasets[37]. A C site was determined as confidently methylated if, for both bisulfite sequencing datasets, 95% of reads were methylated with at least 10× coverage. On the other hand, a C site was considered confidently unmodified if, for both bisulfite sequencing datasets, at most 1% of reads were methylated with at least 10× coverage. Such analysis covered 3367/3367 canonical C-containing DNA 6mers and 3950/6144 single-5mC-containing DNA 6mers.

*Selecting nanopore sequencing reads.* We then ran signalAlign with reads reported in the nanopore consortium NA12878 cell line native genomic DNA datasets[38] covering the above-mentioned prediction sites. Considering the computational complexity of signalAlign, we performed the following filtering steps to use the fewest reads to cover the most kmers. First, we calculated read-level kmer coverage. For example, the center 5mC site of DNA read CAGAT(**5mC**)ACAGA was selected for signalAlign prediction. 6mers CAGAT(**5mC**), AGAT(**5mC**)A, GAT(**5mC**)AC, AT(**5mC**)ACA, T(**5mC**)ACAG, and (**5mC**)ACAGA span such 5mC site and therefore are considered as being covered. Based on such read-level kmer coverage, we iteratively selected reads that covered the least frequently

covered kmers. Thus, building a read set that covers as many kmers as possible as often as possible with the fewest number of reads. We included two biological replicates of NA12878 cell line native genomic DNA-sequencing experiments (FAB39088 and FAF01169) in the C/5mC predictive analysis. For such analysis, our final FAB39088 set contained 1706 reads, which covered 2625/3367 C-only DNA 6mers with an average 61.52× coverage as negative control and 3105/3950 possible single-5mC DNA 6mers with an average 5.01× coverage. The final FAF01169 set contained 1396 reads, which covered 2610/3367 C-only DNA 6mers with an average 63.26× coverage as negative control and 3140/3950 single-5mC DNA 6mers with an average 4.76× coverage. Combining the two sets, in total 2792/3367 C-only DNA 6mers were covered with an average 58.49× coverage and 3481/3950 single-5mC DNA 6mers were covered with an average 4.38× coverage.

*Performing signalAlign prediction*. Based on the selected prediction sites and nanopore sequencing reads as described above, per-read per-site predictive analysis was performed by signalAlign. The signalAlign analysis was performed with default parameters, except for internal read-level quality filtering. Such quality filtering removes reads with poor kmer-to-ionic current correspondence. During signalAlign analysis, kmer-to-ionic current correspondence probability matrices (event tables) are first generated. Based on such event tables, signalAlign will remove reads with low average probabilities ($<10^{-5}$). In addition, reads with >50 consecutive ionic current signals that cannot be corresponded to kmers (probability equals 0) will be discarded. Considering that the event table generation is based on the provided kmer model, after the above-mentioned default quality filtering, the number of remaining reads varies when different kmer models are supplied during predictive analysis. To ensure the statistical soundness, we deactivate the default quality filtering, such that reads to be analyzed by different supplied kmer models will be the same.

*Performing megalodon prediction*. We also performed predictive analysis using the deep learning-based modification calling algorithm megalodon (https://github.com/nanoporetech/megalodon) as an additional baseline control. The megalodon (version 2.3.1) analysis was performed with tags "<fast5>–outputs mod_mappings mods --reference <reference>--processes 1 --overwrite --guppy-server-path guppy_basecall_server --output-directory <output dir>--guppy-timeout 1000 --guppy-concurrent-reads 1 --guppy-params'--num_callers 7--cpu_threads_per_caller 10--chunks_per_runner 100'."

Considering the extraordinary performance of megalodon (Supplementary Fig. 5), we further used megalodon predictions as additional ground truth for the C/5mC status for every nanopore sequencing read at every prediction site. Please see Supplementary Note 3 for more information.

*Quantifying predictive accuracy*. signalAlign quantifies the probability of being C or 5mC for every prediction. We used probability threshold 0.7 to ensure only confident predictions were included in predictive accuracy quantification. Together with the megalodon 5mC calling results, we further created confusion matrices (2 × 2 for 5mC predictive analysis with 5mC as "positive" class and C as "negative" class) to quantify predictive accuracy. Specifically, we calculated the true-positive rate, true-negative rate, positive predictive value, negative predictive value, F1 score (F1), and BA as predictive accuracy quantifications. BA was presented in Fig. 1E as representative quantification and the full predictive performance can be found in Supplementary Fig. 5.

### *Escherichia coli* 16S rRNA 2mG-site analysis
*Ionic current signal distributions*. We first downloaded the nanopore sequencing fast5 reads of *E. coli* 16S rRNA nanopore sequencing reads reported in ref. [14]. We then performed nanopolish extract analysis[3,23] to retrieve the fastq records, with tags "-v -r -q -t template." The fastq records were then aligned using minimap2 (2.16-r922)[39] with flags "-ax map-ont," further sorted and indexed by samtools (1.12)[40]. Per-read event tables were generated using nanopolish eventalign with flag "--scale-events," by taking fast5 reads, alignment files, and retrieved fastq records as described above. The yielded event tables contain RNA 5mer sequences and corresponding ionic current signals. We then quantified the distributions of RNA 5mer ionic current signals.

*Predictive analysis*. We also performed predictive analysis for the {A, 6mA, T, G, 2mG, C} RNA 5mer model described in "predicting modification-containing kmers." Specifically, we tested whether the predicted RNA 5mer model could be used to correctly identify the position of 1206 2mG site, as well as three nearby G sites (positions 851, 1221, and 1386) in *E. coli* 16S rRNA (see https://www.ncbi.nlm.nih.gov/nuccore/J01859 for details). We thus ran signalAlign with nanopore sequencing reads reported in ref. [14], following the same steps as described in "human genome C/5mC-status predictive analysis." We also used probability threshold 0.7 to select confident predictions.

### Kmer models
Canonical DNA 6mer and RNA 5mer models are available at: https://github.com/nanoporetech/kmer_models. The nanopolish 5mC-containing DNA 6mer model is available at: https://github.com/nanoporetech/nanopolish/tree/master/etc/r9-models. The GSE124309 model, which contains the union of

{A, U, C, G} and {6mA, U, G, C} RNA 5mers, was constructed by the following steps. We first downloaded the nanopore sequencing fast5 reads of modified and non-modified "curlcake constructs" replicate 1 with GEO accession code GSE124309[13]. We then performed nanopolish extract analysis[3,23] to retrieve the fastq records, with tags "-v -r -q -t template." The fastq records were then aligned using minimap2 (2.16-r922)[39] with flags "-ax map-ont," further sorted and indexed by samtools (1.12)[40]. Per-read event tables were generated using nanopolish (0.11.1) eventalign with flag "--scale-events," by taking fast5 reads, alignment files, and retrieved fastq records as described above. The yielded event tables contain RNA 5mer sequences and corresponding ionic current signals. For every RNA 5mer, we averaged ionic current signals of all instances recorded in the event tables to build the GSE124309 model. Please note that for more recent nanopore sequencing chemistries, e.g., R10 where ONT kmer models are no longer available, empirical kmer models could be trained instead as above-mentioned. Please see Supplementary Note 4 for details.

**Reporting summary**. Further information on research design is available in the Nature Research Reporting Summary linked to this article.

### Data availability
The FAB39088 and FAF01169 NA12878 cell line native genomic DNA nanopore sequencing datasets were downloaded from https://github.com/nanopore-wgs-consortium/NA12878/blob/master/Genome.md. The two independent NA12878 bisulfite datasets were downloaded from https://www.encodeproject.org/experiments/ENCSR890UQO/. The *E. coli* 16S rRNA nanopore sequencing dataset was reported by Smith et al.[14]. The nanopore sequencing dataset used to construct the GSE124309 model is available at GEO under the accession code GSE124309[13].

### Code availability
Codes for constructing, training, and running the deep learning framework are available at https://github.com/ioannisa92/Nanopore_modification_inference[41]. Codes for nanopore sequencing data analysis are available at https://github.com/adbailey4/functional_model_analysis[42]. Specifically, we adapted the original nanopolish (0.11.1) for our analysis. The adapted nanopolish is available at https://github.com/adbailey4/nanopolish[43]. Codes for reproducing all figures are available upon request to the corresponding authors.

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

## Acknowledgements
Research reported in this publication was supported by the National Institutes of Health under Award Numbers R01-HG010053-02, U01HG010961, U41HG010972, R01HG010485, 2U41HG007234, 5U54HG007990, 5T32HG008345-04, and U01HL137183. The content is solely the responsibility of the authors and does not necessarily represent the official views of the National Institutes of Health. The authors would thank Jordan Eizenga, Dr. Jonas Sibbesen, Dr. Mark Akeson, and Dr. Miten Jain for critical insight and help with drafting the manuscript.

## Author contributions
H.D. conceived the idea. I.A. performed deep learning framework modeling, optimization, and analysis. A.D.B. and H.D. performed the nanopore sequencing data analysis. H.D., J.S., and B.P. supervised the project. All authors prepared the manuscript.

## Competing interests
The authors declare no competing interests.
