## [Peer Review File · Nature Communications]

Towards Inferring Nanopore Sequencing Ionic Currents from Nucleotide Chemical StructuresEditorial Note: This manuscript has been previously reviewed at another journal that is not operating a transparent peer review scheme. This document only contains reviewer comments and rebuttal letters for versions considered at Nature Communications.

Reviewers' Comments:

Reviewer #4:

Remarks to the Author:

Summary:

Many of Reviewer #2's concerns seem to stem from a general disagreement over the scope and focus of the manuscript. Given the number of less common base modifications that could potentially be reflected in nanopore signal, as well as the difficulty of generating these kinds of training sets, having a way of generalizing basecallers/polishers to new modifications in a data-efficient manner seems quite useful. The authors make it clear that this is the motivation and focus of their method, which is decidedly not the same as developing the best possible 5mC caller or one that works on a limited set of modifications. I think the strength of this manuscript is as a proof-of-concept for using GCN layers to learn about and generalize between similar bases and modifications. The natural extension of this work would be to incorporate it into an end-to-end deep learning tool for modification calling, which the authors themselves suggest in the discussion.

Response to Reviewer #2's specific comments:

4. I agree with the authors' response here.

5. No comment.

6. Given the other study cited here, it seems that reproducible k-mer models can be generated from modest amounts of sequencing data. However, I think there are valid concerns about the future of k-mer models for nanopore sequencing. As the comparison in Fig S5 shows, Megalodon far outperforms existing k-mer models for 5mC detection, and it relies on a neural network basecaller trained explicitly on a fifth base. That being said, the GCN approach of the authors should hopefully generalize to some of these deep learning models, though I accept that it is outside the scope of this manuscript.

7. The explanation given by the authors makes sense to me. Given the previous point about training new k-mer models, it should also be relatively straightforward to train a 5mer model of DNA which would allow for a direct comparison to the 5mer RNA model.

8. No comment.

9. I agree with the authors that de novo prediction is one of the main strengths of this approach, and agree with including the de novo prediction panel in the same figure.

10. I think it's quite reasonable to use signalAlign for the actual predictions while using the k-mer model from nanopolish. Labeling the baseline in figures 1 and S5 from "nanopolish" to something like "nanopolish model" might help avoid Reviewer #2's confusion. As it stands currently, it contrasts with the megalodon baseline, where "megalodon" just refers to running megalodon out of the box (without signalAlign).

The use of balanced accuracy as a metric also seems appropriate, though it could be motivated more

clearly by mentioning the class imbalance in this dataset (e.g. what minority of positions are methylated in their dataset), as well as including a definition of balanced accuracy in the methods.

11. The authors construct their ground truth C/5mC set by looking at positions present in two different bisulfite sequencing experiments run on the same cell line but not the same sample. This seems reasonable to me, though I agree it would've been cleaner to have the same sample.

12. I think the authors' approach is reasonable and agree with their response.

13. I agree with the authors' response.

14. While the feature vectors output by the GCN are clearly central to this work, I do share Reviewer #2's concern that the correlation heat maps are difficult to interpret. I wonder if there are other ways of visualizing the underlying feature vectors more directly besides just correlation? For instance, visualizing each feature vector (per atom or averaged over a chemical moiety / group) as a point in lower dimensional space. This could visualize, for example, the chemical encoding of methyl groups across k-mers.

15. The author have added more examples to the SI, which seem to partially address Reviewer #2's concern. Beyond individual examples, perhaps the authors could break down the predictive accuracy by sequence context, since that seems to be a concern of Reviewer #2.

16. I'm not sure exactly what Reviewer 2 is proposing in looking at "expected cluster formation", unless they are referring to clusters of the feature vectors themselves, as I mention in #14.

I understand there has been some work using molecular mechanics to predict ionic current in nanopores, but given the sensitivity of the current to pore version (e.g. R10 vs R9.4) and the fact that ONT does not release the identity of the pore protein, getting any sort of agreement between a generic nanopore simulation and a model trained on ONT data seems extremely difficult and outside the scope of this work.

17. Beyond the point already made in #14, it seems the authors have addressed the reviewer's concern with the inclusion of several example k-mer structures in the SI.

18. No comment.

19. No comment.

20. No comment.

Reviewer #4 (Remarks to the Author):

Summary:

Many of Reviewer #2's concerns seem to stem from a general disagreement over the scope and focus of the manuscript. Given the number of less common base modifications that could potentially be reflected in nanopore signal, as well as the difficulty of generating these kinds of training sets, having a way of generalizing basecallers/polishers to new modifications in a data-efficient manner seems quite useful. The authors make it clear that this is the motivation and focus of their method, which is decidedly not the same as developing the best possible 5mC caller or one that works on a limited set of modifications. I think the strength of this manuscript is as a proof-of-concept for using GCN layers to learn about and generalize between similar bases and modifications. The natural extension of this work would be to incorporate it into an end-to-end deep learning tool for modification calling, which the authors themselves suggest in the discussion.

Response to Reviewer #2's specific comments:

14. While the feature vectors output by the GCN are clearly central to this work, I do share Reviewer #2's concern that the correlation heat maps are difficult to interpret. I wonder if there are other ways of visualizing the underlying feature vectors more directly besides just correlation? For instance, visualizing each feature vector (per atom or averaged over a chemical moiety / group) as a point in lower dimensional space. This could visualize, for example, the chemical encoding of methyl groups across k-mers.

We thank the reviewer for the comment. In the updated Figure 2, we included tSNE dimension reduction plot, in which "each feature vector as a point" (panel B and E). As shown in the two panels, atoms under the same chemical context cluster together, e.g. T-methyl group carbon atom #38 and 5mC-methyl group carbon atom #58 in GT(5mC)AGA.

15. The author have added more examples to the SI, which seem to partially address Reviewer #2's concern. Beyond individual examples, perhaps the authors could break down the predictive accuracy by sequence context, since that seems to be a concern of Reviewer #2.

We thank the reviewer for the comment. In the updated Supplementary Note 4, we included context-specific predictive accuracy analysis.